# Hip Spacers with a Metal-on-Cement Articulation Did Not Show Significant Surface Alterations of the Metal Femoral Head in Two-Stage Revision for Periprosthetic Joint Infection

**DOI:** 10.3390/ma13173882

**Published:** 2020-09-02

**Authors:** Andre Lunz, Robert Sonntag, J. Philippe Kretzer, Sebastian Jaeger, Therese Bormann, Marcus R. Streit, Nicholas A. Beckmann, Burkhard Lehner, Georg W. Omlor

**Affiliations:** 1Clinic for Orthopedics and Trauma Surgery, Center for Orthopedics, Trauma Surgery and Spinal Cord Injury, Heidelberg University Hospital, Schlierbacher Landstrasse 200a, 69118 Heidelberg, Germany; streitmarcus@gmail.com (M.R.S.); nicholas.beckmann@med.uni-heidelberg.de (N.A.B.); burkhard.lehner@med.uni-heidelberg.de (B.L.); georg.omlor@med.uni-heidelberg.de (G.W.O.); 2Laboratory of Biomechanics and Implant Research, Clinic for Orthopedics and Trauma Surgery, Heidelberg University Hospital, Schlierbacher Landstrasse 200a, 69118 Heidelberg, Germany; Robert.Sonntag@med.uni-heidelberg.de (R.S.); philippe.kretzer@med.uni-heidelberg.de (J.P.K.); Sebastian.Jaeger@med.uni-heidelberg.de (S.J.); therese.bormann@med.uni-heidelberg.de (T.B.)

**Keywords:** periprosthetic joint infection, cement spacer, articulating spacer, hip spacer, two-stage revision, surface alteration, surface roughness, third-body wear, zirconium oxide particles, metal-on-cement articulation

## Abstract

Two-stage revision is considered the gold standard treatment for chronic periprosthetic joint infection (PJI). During the interim period, between explantation of the infected hip endoprosthesis and revision arthroplasty, individually formed articulating hip spacers made of polymethylmethacrylate (PMMA) bone cement can be used to provide better soft tissue preservation, local antibiotic release, and improved postoperative mobilization. If effective prevention from luxation is achieved, hip function and hence overall patient satisfaction is improved. Zirconium oxide particles inside conventional PMMA bone cement, however, are known to enhance third-body wear, which may cause alterations of the metal head in the articulating spacer and hence become a serious risk for the patient. Therefore, the aim of our study was to analyze whether the articular surface of cobalt-chrome (CoCr) femoral heads is significantly altered in the setting of a metal-on-cement articulation during the interim period of two-stage revision for PJI. We analyzed a consecutive series of 23 spacer cases and compared them with femoral heads from two series of conventional hip arthroplasty revisions with metal-on-polyethylene articulations and different time intervals in situ. To investigate metallic wear, the femoral heads were thoroughly examined, and their surface roughness was measured and analyzed. We found no significant differences between the two conventional hip arthroplasty groups, despite their very different times in situ. Furthermore, the individually different times in situ within the spacer group had no significant impact on surface roughness, either. Compared with the spacer group, the surface roughness of the metal femoral heads from both conventional hip arthroplasty groups were even higher. Within the spacer group, roughness parameters did not show significant differences regarding the five predefined locations on the metal head. We conclude that metal-on-cement articulations do not cause enhanced surface alterations of the metal femoral head and hence do not limit the application in articulating hip spacers in the setting of two-stage revision for PJI.

## 1. Introduction

Periprosthetic joint infection (PJI) is a dreaded complication of total joint replacement that brings numerous challenges to the affected patients and the orthopedic surgeons. One-stage revision has recently gained more support because of its good clinical results and numerous advantages in carefully selected patients. A widely accepted consensus was reached on several strict exclusion criteria such as unknown or multiresistant organism-based infections, relevant comorbidities, sinus tract, and other relevant soft tissue complications [1,2,3]. Therefore, two-stage revision is still widely considered the gold standard treatment in the setting of chronic and complex periprosthetic joint infections. Insall et al. first described this technique in 1983 [4]. Since then, numerous studies have reported that two-stage revision has been used successfully and can result in infection eradication rates beyond 90% [5,6,7]. The treatment regimen includes aggressive debridement, taking of numerous samples for histopathological and microbiological analysis, and removal of all implants in the first stage. To avoid a Girdlestone hip, where the implant or joint resection is conducted without replacement, an antibiotic loaded polymethylmethacrylate (PMMA) cement spacer can be implanted for the interim period. The advantage of a spacing device is not only that the soft tissue envelope will be better preserved at the time of reimplantation, but also that articulating spacers help prevent arthrofibrosis, limb shortening, and osteolysis, leading to an overall improved outcome [7,8,9]. Mobile hip spacers create an articulation between the femoral head and acetabulum, similar to the articulation in conventional total hip arthroplasty, resulting in improved hip function during the interim period and consequently after final reimplantation [10,11,12]. Several techniques for constructing mobile spacers have been described, [12,13,14] providing a metal-on-polyethylene articulation with the major disadvantage of using a polyethylene (PE) liner in the setting of infection, leading to an increased risk of biofilm formation on its surface. An articulating spacer technique using an individually formed metal-on-cement articulation can be used as an alternative. Here, a custom-made antibiotic-loaded cemented acetabular socket is formed to provide a congruent articulating surface with a conventional metal femoral head. A deliberately loosely cemented unsophisticated femoral stem serves to increase the strength of the construct and at the same time reduces the risk of periprosthetic fracture. In practice, this spacer technique allows early patient mobilization with a good range of motion and at least partial weight-bearing activity directly from postoperative day one [own unpublished data].

The aim of the current study was to analyze whether the articular surface of the metal (CoCr) femoral head is altered in the setting of a metal-on-cement articulation. This major concern arises because PMMA bone cement contains hard zirconium-oxide-particles, which are known to be responsible for enhanced third-body wear after total joint replacement [15,16,17,18]. Since we have had good clinical experience with this described articulating spacer technique, our aim was to evaluate its biomechanical usability. We hypothesized that surface roughness of the femoral heads from spacers would not increase significantly compared to conventional metal-on-polyethylene articulations, allowing the conclusion that no relevant wear has occurred. We therefore analyzed and compared the surface roughness of metal femoral heads from articulating cement spacers with metal femoral heads from conventional hip arthroplasty revisions with metal-on-polyethylene articulations.

## 2. Materials and Methods 

All revision procedures were performed at the same university institution. First, a consecutive series, from November 2017 to August 2019, of 23 articulating hip spacer cases was analyzed. All spacers had a mobile metal-on-cement articulation and were implanted for the interim period in the setting of PJI. Two series of conventional hip arthroplasty revision cases with metal-on-polyethylene articulations with heterogeneous in situ time intervals were analyzed and compared as a reference. All implants were collected on a standardized base. All operations were performed only for medical reasons by specialized surgeons at our certified over-regional joint arthroplasty center. All patients have given written consent for their retrieved implants to be used for different research purposes. This study was approved by the local Medical Ethics Committee (S-091/2018).

### 2.1. Clinical Data

The study included 23 femoral head components from performed spacer explantations in the setting of two-stage revision for PJI (Table 1, No. 1–23). The interim period between explantation and reimplantation was variable and based on clinical, radiological, and laboratory evidence that infection had been overcome. In 6 cases, the collected intraoperative samples during explantation surgery only showed histopathological evidence of infection without microbiological detection of any microorganisms and in 3 cases more than one bacterium was found. In another case a fungus was established as the cause of infection. During second-stage surgery the microbiologic samples of 4 cases stayed positive, as reported in Table 2.

The mean interim period and hence mean time of spacers in situ was 79 (14–253) days. The spacer components came from 13 men and 10 women. Mean patient age was 70 years at the time of spacer implantation. As this spacer technique allows early patient mobilization, we tried to standardize weight bearing as much as possible by giving the same instructions to all patients and by using the help of a specially trained physical therapist performing standardized mobilization protocols to all patients.

In all 23 cases, the metal-on-cement articulation was constructed using a 28 or 32 mm CoCr femoral head (S-2XL, DePuy Synthes, West Chester, PA, USA) with a corresponding femoral stem (Weber Stem CS/CM/SM, Zimmer, Warsaw, IN, USA) and a custom-made acetabular socket formed out of 40–80 g of antibiotic-loaded PMMA cement (Palacos cement, Heraeus, Hanau, Germany) with 3 g Vancomycin powder per 40 g of cement, as shown in Figure 1 (own unpublished data).

After a sufficient number of samples are taken and radical debridement is completed, the spacer is constructed as follows: First, the acetabular shell is loosely formed by hand and put as dough into the acetabular groove. Before PMMA polymerization has occurred, surplus cement is removed at the acetabular edges and an articulation groove is formed in the middle of the acetabular spacer using a plunger with a slightly larger diameter of its head than the original metal head of the later articulation. To form a smooth groove, the plunger is continuously rotated and moved until PMMA polymerization has finished and the acetabular component is fixed. Later, this allows free movement of the metal head inside the cement groove. Then, as the second step, the femoral component is built by applying cement on the metal stem. To facilitate later removal, no cement is applied around the tip of the stem. After a couple of minutes, the cement becomes more solid and the stem is carefully pushed into the femur. To avoid cement penetration into the bone, the stem is continuously moved back and forwards until PMMA polymerization has completed. From our experience, this “deliberately loose cementing technique” provides sufficient fixation in the bone and allows easy removal in the second stage. After both spacer components (acetabular and femoral) are in place, a metal head is put on the stem and the reduction maneuver is carefully performed. Finally, it is extremely important to check for sufficient hip stability in flexion, extension, and rotation to minimize the risk of spacer luxation. If the articulation of the metal head inside the cement groove is not stable enough (e.g., it can be easily luxated), the spacer cannot be retained and must be replaced by a new one.

To allow comparisons with metal-on-polyethylene articulations in short- and long-term settings, the study also included two series from conventional hip arthroplasty revisions with a CoCr femoral head and a metal-on-polyethylene articulation (Table 1, No. A–C and No. D–L), with short and long implant periods in situ. All revision surgeries were carried out due to confirmed or highly suspected PJI. The mean period in situ was 299 (89–459) days and 11 (5–17) years in the short and in the long implant period group, respectively. Mean age of the patients at explantation was 72 and 78 years, respectively, as reported in Table 1.

### 2.2. Femoral Head Analysis

To determine alterations of the metal head as a sign for possible metallic wear, the femoral heads were analyzed for surface roughness and deviations in shape caused by in situ wear. A tactile roughness measurement instrument (Perthomether M2 profilometer; Mahr, Göttingen, Germany) was used to characterize the surface of the femoral heads with 12 nm of accuracy at 5 predefined locations on the implants, as shown in Figure 2. Care was taken to avoid measuring areas that most probably had been damaged during revision surgery and not by in situ wear. All measurements were performed according to ISO 4287 [19] with a scanning length of 1.25 mm. The parameters, which were used to describe and interpret the surface alterations of the femoral heads, are described in Table 3. The common parameter average roughness (Ra) and average maximum profile height (Rz) provide a general basis to evaluate and interpret surface topographies. Since the definition of Ra is based on arithmetic averaging and Rz on an average of five maximum heights, these values are by definition unable to distinguish between peaks and valleys. Thus, the core roughness depth (Rk) that has been established to characterize load-carrying surfaces has been used [20,21].

The metallic surface of the CoCr femoral heads was further examined, and light optical images were taken with a digital microscope (VHX-5000, Keyence, Osaka, Japan) at a magnification ×200.

### 2.3. Statistical Analysis

All descriptive data is presented as the arithmetic mean and standard deviation. The Kruskal–Wallis test was used to compare the means between the groups. Pearson correlations were used to determine statistical correlations between clinical parameters (time in situ, patient age, femoral head size) and roughness parameters. The level of significance was set at *p* < 0.05 for all statistical tests. The statistical analyses were performed using SPSS software (version 25.0; SPSS Inc, Chicago, IL, USA).

## 3. Results

### 3.1. Roughness Measurements

First, differences in the spacer group between roughness measurements of the five investigated locations on the femoral head (Figure 2) were analyzed. No significant statistical differences were found, as shown in Table 4.

The surface roughness parameters between the spacer group and the two conventional hip arthroplasty revision groups were compared. Since significant differences were found, a further pairwise comparison was conducted. No significant differences were shown regarding the two conventional hip arthroplasty groups and their different implant times in situ. Compared with the spacer group, the surface roughness of the femoral heads in the metal-on-polyethylene bearings was significantly increased with regard to all measured roughness parameters, except for Ra, which was also higher, but without reaching significance. The results are summarized in Table 5 and Table 6.

Furthermore, light optical images were taken of the metallic articular surface using a magnification × 200. An example of this is shown in Figure 3, in which surface alterations in the spacer group are markedly less prominent in comparison to both conventional hip arthroplasty groups. A conventional hip endoprosthesis with a metal-on-metal articulation was used as a reference and showed even more surface roughness evident as more scratches and grooves than all three groups.

### 3.2. Correlations to Clinical Data

We further looked for correlations between roughness parameters within the spacer group and available clinical parameters, such as patient age, time of the implant in situ, and femoral head size. Again, no significant statistical correlations were found (Table 7).

## 4. Discussion

This study analyzed and compared the surface roughness of metal femoral heads from explanted articulating spacers and conventional hip arthroplasty revisions. All surgery was performed because of confirmed or highly suspected PJI. We assumed that a higher surface roughness would most likely lead to more metallic and cement wear. All analyzed articulating spacers had metal-on-cement articulations, which could be prone to enhanced third-body wear because of hard zirconium oxide particles in the PMMA bone cement. The goal of our study was to evaluate biomechanical usability against the background of this assumption.

Interestingly, no significant differences between the two conventional hip arthroplasty groups were found, despite their very different times in situ. Compared with the spacer group, the surface roughness of both conventional hip arthroplasty groups was significantly higher. This allows the conclusion that in the setting of two-stage revision for PJI, metal-on-cement articulations in articulating spacers show less surface modification on the metallic head than metal-on-polyethylene articulations in conventional hip arthroplasties. The amplified light optical images of the metal femoral heads confirm these results. Femoral heads from the spacer group show unambiguously fewer surface alterations than both conventional hip arthroplasty groups. The problem of enhanced third-body wear in joint replacement, known to be caused by zirconium oxide particles in PMMA bone cement, seems to be neglectable in the setting of two-stage revision for PJI. This is probably because of the rather short interim period and thus time in situ.

Wear is also thought to be influenced by the patient’s activity level and thus implant exposure in situ. The articulating spacer system allows hip joint movement (usual range of motion: extension/flexion: 0/0/90°) and partial weight bearing (usually 20 kg body weight). This is an improvement when compared to a Girdlestone hip, but remains more restricted than in patients who received conventional total hip replacement. Limited activity and partial weight bearing in the spacer group could also be an important reason for the differences in surface roughness compared with the conventional hip arthroplasty groups. Nevertheless, these findings justify the described spacer technique using a metal-on-cement articulation despite the enhanced risk for early (low level) third-body wear. 

The roughness parameters within the spacer group showed no significant differences regarding the five analyzed locations on the femoral head. In contrast, patients suffering from osteoarthritis have very well described locations on the femoral head, where increased cartilage wear is found. This implies that the measured surface roughness, and therefore wear, was most probably too minute to permit significant differences in the setting of articulating spacers. 

No significant correlations between clinical data (implant time in situ, patient age, or femoral head size) and roughness parameters within the spacer group were found. As discussed, one would assume that longer times in situ would enhance implant exposure and thus create increased surface roughness and wear. Once again, the uniformly low patient activity level during the interim period was probably responsible for this lack of significance. Furthermore, the period of time of all spacers in situ was well under a year and thus possibly not long enough to create significant surface alterations. 

To summarize, CoCr femoral heads from articulating spacers showed decreased surface roughness parameters in comparison to retrieved metal femoral heads from conventional hip arthroplasties. The small amount of surface alterations on the CoCr femoral heads allows the assumption that metallic wear is considerably low. These findings implicate that metal-on-cement articulations in interim spacers are biomechanically usable in the setting of two-stage revision for PJI.

## Figures and Tables

**Figure 1 materials-13-03882-f001:**
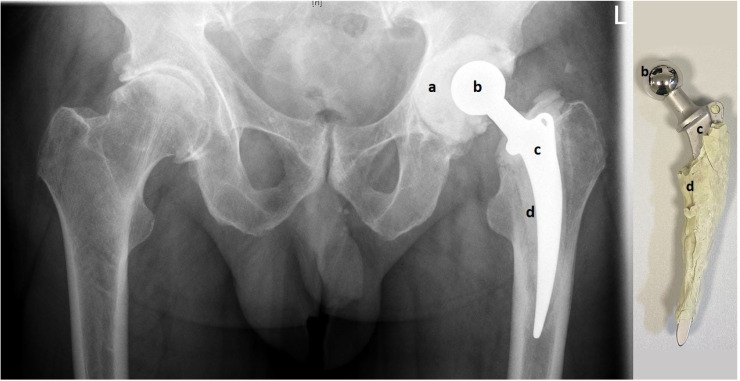
Pelvic X-ray with articulating spacer in situ (left) and the same spacer after explantation (right). The custom-made acetabular socket (**a**) is made of antibiotic-loaded PMMA (polymethylmethacrylate) bone cement and creates the metal-on-cement articulation with the CoCr femoral head (**b**). PMMA bone cement (**d**) is also used to fix the femoral stem into the femur (**c**).

**Figure 2 materials-13-03882-f002:**
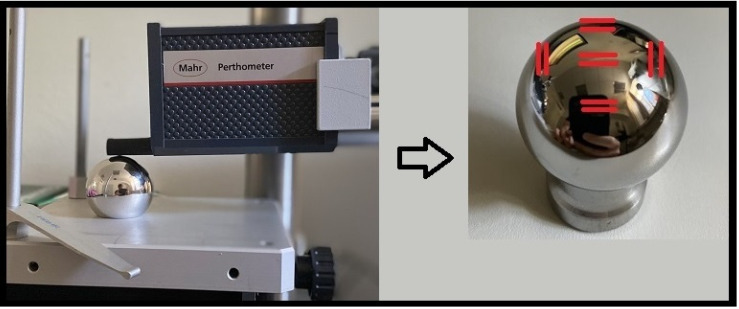
Shows the 5 predefined locations where the roughness measurements were performed.

**Figure 3 materials-13-03882-f003:**
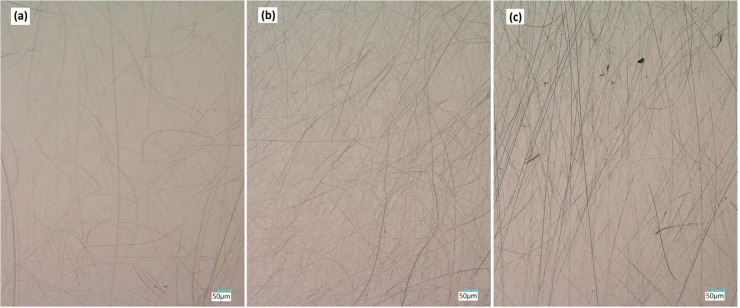
Representative examples of light optical images of the metal femoral heads at a magnification ×200. (**a**) Spacer with metal-on-cement articulation, (**b**) conventional hip arthroplasty with metal-on-polyethylene articulation, and (**c**) conventional hip arthroplasty with metal-on-metal articulation.

**Table 1 materials-13-03882-t001:** Clinical data of the 3 groups: articulating spacer patients (No. 1–23), conventional hip arthroplasty patients with short implant time in situ (No. A–C), and long time in situ (No. D–L).

No.	Patient Age	Patient Gender	Femoral Head Size	Time In Situ
	(years)	(m = male, f = female)	(mm)	(d = days, y = years)
1	76	m	32	44 d
2	67	f	32	17 d
3	68	m	32	86 d
4	78	m	28	84 d
5	67	m	28	14 d
6	57	m	28	71 d
7	54	f	28	28 d
8	69	f	28	140 d
9	61	m	28	113 d
10	82	m	28	117 d
11	63	f	28	130 d
12	64	f	32	68 d
13	65	m	28	253 d
14	69	f	28	51 d
15	58	f	28	112 d
16	68	m	28	77 d
17	69	f	28	76 d
18	75	m	28	35 d
19	65	f	28	26 d
20	81	f	28	84 d
21	78	m	28	62 d
22	78	m	28	57 d
23	75	m	28	70 d
**Mean (Range)**	**69 (54–82)**	**m = 13, f = 10**	**28 mm = 19, 32 mm = 4**	**79 d (14–253)**
A	71	m	32	350 d
B	67	f	32	459 d
C	77	m	32	89 d
**Mean (Range)**	**72 (67–77)**	**m = 2, f = 1**	**32 mm = 3**	**299 d (89–459)**
D	77	m	32	9 y
E	84	m	32	15 y
F	78	m	28	10 y
G	74	f	28	7 y
H	74 76	f	28	15 y
I	76	f	28	17 y
J	85	m	32	5 y
K	77	m	28	13 y
L	76	m	32	8 y
**Mean (Range)**	**78 (74–85)**	**m = 6, f = 3**	**28 mm = 5, 32 mm = 4**	**11 years (5–17)**

**Table 2 materials-13-03882-t002:** Microbiological results from collected intraoperative samples during a) first- and b) second-stage surgery in the articulating spacer group.

		(a) First-Stage Surgery (Explantation)		(b) Second-Stage Surgery (Reimplantation)	
Microorganism		Number	Percentage (%)	Number	Percentage (%)
**None**		**6**	**26.1**	**19**	**82.6**
**Coagulase-positive staphylococci**		**2**	**8.7**		
	Staph. aureus	2	8.7		
**Coagulase-negative staphylococci**		**11**	**47.8**	**3**	**13**
	Staph. lugdunensis	2	8.7		
	Staph. capitis	2	8.7		
	Staph. epidermidis	7	30.4	3	13
**Other**		**4**	**17.4**		
	Bacillus spec.	1	4.3		
	Cutibacterium acnes	3	13		
**Fungal infection**		**1**	**4.3**	**1**	**4.3**
	Candida albicans	1	4.3	1	4.3
**Polymicrobial infections**		**3**	**13**		

**Table 3 materials-13-03882-t003:** This table summarizes the used surface characteristics parameters [19,20,21].

Parameter	Definition	Description	Standard
*Ra*	Arithmetic average profile roughness	Arithmetic average of the absolute values of the roughness profile ordinates	EN ISO 4287
*Rz*	Average maximum height of the profile	Average of the 5 highest peaks and 5 deepest valleys in the profile	EN ISO 4287
*RSm*	Mean line peak spacing	The mean spacing between the profile peaks over the sampling length	EN ISO 4287
*Rp*	Maximum peak height of the profile	The height of the highest peak above the mean line within the sampling length	EN ISO 4287
*Rk*	Core roughness depth	The depth of the roughness core profile within the evaluation length, excluding the height of protruding peaks and deep valleys. Rk is obtained from the material ratio curve (Abbott curve)	EN ISO 13565

**Table 4 materials-13-03882-t004:** Analysis of roughness parameters in the spacer group at the 5 predefined locations. Statistical significance is assumed for *p* < 0.05 and was not found.

	Roughness Parameters in µm (Mean ± Standard Deviation)				
Predefined Measurement Places on the Femoral Heads	Ra	Rz	RSm	Rp	Rk
No. 1–2	0.054 ± 0.01	0.230 ± 0.07	255.5 ± 157.3	0.200 ± 0.08	0.064 ± 0.02
No. 3–4	0.052 ± 0.01	0.222 ± 0.11	372.8 ± 301.8	0.195 ± 0.09	0.063 ± 0.02
No. 5–6	0.046 ± 0.01	0.208 ± 0.07	280.0 ± 258.8	0.175 ± 0.06	0.058 ± 0.01
No. 7–8	0.050 ± 0.02	0.221 ± 0.15	177.8 ± 118.5	0.175 ± 0.09	0.066 ± 0.03
No. 9–10	0.048 ± 0.01	0.212 ± 0.10	286.6 ± 203.5	0.176 ± 0.07	0.062 ± 0.02
***p* value**	**0.153**	**0.541**	**0.072**	**0.342**	**0.654**

**Table 5 materials-13-03882-t005:** Comparison of roughness parameters between the spacer group and two conventional hip arthroplasty revision groups (PE-Group Short = short implant time in situ, PE-Group Long = long implant time in situ).

	Roughness Parameters in µm (Mean ± Standard Deviation)				
	Ra	Rz	RSm	Rp	Rk
Spacer Group	0.050 ± 0.01	0.221 ± 0.12	273.2 ± 222.3	0.183 ± 0.08	0.063 ± 0.02
PE-Short Group	0.061 ± 0.03	0.312 ± 0.14	189.5 ± 95.7	0.233 ± 0.06	0.084 ± 0.04
PE-Long Group	0.056 ± 0.03	0.355 ± 0.19	128.9 ± 106.1	0.244 ± 0.13	0.089 ± 0.09
***p*-value**	**0.062**	**<0.01 ***	**<0.01 ***	**<0.01 ***	**<0.01 ***

*: The level of significance was set at *p* < 0.05

**Table 6 materials-13-03882-t006:** Pairwise Comparison between all 3 Groups (Spacer, PE-Short, PE-Long).

	Rz	RSm	Rp	Rk
Pairwise Comparison	*p* value	*p* value	*p* value	*p* value
Spacer vs. PE-Short	<0.01 *	<0.01 *	<0.01 *	<0.01 *
Spacer vs. PE-Long	<0.01 *	<0.01 *	<0.01 *	<0.01 *
PE-Short vs. PE-Long	0.532	0.234	0.756	0.811

*: The level of significance was set at *p* < 0.05

**Table 7 materials-13-03882-t007:** Calculated p values for statistical correlations between roughness parameters and clinical parameters in the spacer group. Statistical significance is assumed for *p* < 0.05 and was not found.

	Roughness Parameters				
Clinical Parameters	Ra	Rz	RSm	Rp	Rk
Time In Situ	0.507	0.767	0.658	0.932	0.816
Patient Age	0.864	0.267	0.748	0.863	0.254
Femoral Head Size	0.088	0.087	0.811	0.469	0.052

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
