# Peer review of "Hip Spacers with a Metal-on-Cement Articulation Did Not Show Significant Surface Alterations of the Metal Femoral Head in Two-Stage Revision for Periprosthetic Joint Infection"

_materials, 2020, doi:10.3390/ma13173882_

Round 1

Reviewer 1 Report

The authors analyze whether the articular surface of cobalt-chrome (CoCr) femoral heads is significantly altered in the setting of a metal-on-cement articulation during the interim period of two-stage revision for PJI. The current study is very interesting, however, I have some concerns.

  1. The cohort is too small to confirm the author's conclusion. So, I would say that the authors should investigate more patients.
  2. I agree with the two-stage revision is useful, but you should discuss about one-stage revision.
  3. Please describe the infected bacteria concretely.

Reviewer 2 Report

An interesting paper. Just a few comments to maybe include in the description to aid surgeon readers:

Please describe how the cement spacer was made ie was the cement put as dough into the acetabulum and the head reduced into it as it was setting or was the acetabulum made on the head outside the acetabulum and allowed to set before reduction. In both cases, was the articulation noted to move freely or a little stiffly and was that different at the time of explantation.

Was any wear or fracture of the cement noted?

How was weight bearing assessed. Giving instructions of '20%' is different to the reality ie some will fully weight bear and some will stay all day in a wheelchair and would that difference make a difference?

One problem with monoblock cement spacers is when the mouth of the acetabulum is smaller than the cavity within, which can make them very difficult to remove at the second stage. Hence a 'beefburger' design has been described to aid removal. Did the authors have any problems with removal?

Round 2

Reviewer 1 Report

The authors answered for concerns and it is suitable for publication.